# The Effect of Place of Residence on Physical Fitness and Adherence to Mediterranean Diet in 3–5-Year-Old Girls and Boys: Urban vs. Rural

**DOI:** 10.3390/nu10121855

**Published:** 2018-12-01

**Authors:** Gema Torres-Luque, Raquel Hernández-García, Enrique Ortega-Toro, Pantelis T. Nikolaidis

**Affiliations:** 1Faculty of Humanities and Science Education, University de Jaén, 23071 Jaén, Spain; gtluque@ujaen.es; 2Faculty of Sport Science, University of Murcia Campus Mare Nostrum, 30720 San Javier, Spain; rhernandezgarcia@um.es; 3Exercise Physiology Laboratory, 184 Nikaia, Greece; pademil@hotmail.com

**Keywords:** Mediterranean diet, preschool, sex, environment

## Abstract

The aim of the present study was to examine the effect of place of residence on physical fitness and adherence to Mediterranean Diet (AMD) in 3–5-year-old children, i.e., whether those who live in urban areas differ than those in rural and whether any difference varied by age. Participants were 363 preschoolers (age 3–5 years old), who performed a series of anthropometric, physical fitness tests and measured their nutritional habits through a 16-item Mediterranean Diet (KIDMED) questionnaire. The main findings of the present study were that (a) boys had better performance in ball bouncing, Medicine Ball Throw Test (MBTT), 25 m sprint, Standing Broad Jump (SBJ), crawling, and shuttle run test (SRT) than girls, and the magnitude of these differences was small; (b) preschoolers in urban residence were taller than those in rural and had better performance in SRT; (c) older preschoolers had larger anthropometric characteristics and better performance than younger preschoolers; (d) the magnitude of the effect of age was larger than the effect of residence; and (e) good AMD was more prevalent in boys than in girls and in 3-year-old participants than in their older peers, but was not related to place of residence. Therefore, these findings identified the need to develop exercise and nutrition intervention programs in preschoolers considering sex, age, and place of residence.

## 1. Introduction

Aspects such as physical fitness and nutritional habits are of vital importance at an early age of human life, and the place of residence and age can be decisive [1,2]. Accordingly, there has been growing interest on the validation of different physical fitness test batteries such as the ALPHA [1] for children between 6 to 12 years old, which highlighted the need for further research in younger children, e.g., 3 to 6 years old. There have been many tests developed for these ages [2], and several test batteries evaluated physical fitness holistically, such as the Movement Assessment Battery for Children-2 [3,4], PREFIT battery [5,6], the Test of Gross Motor Development TGMD-3 [7], and other proposals [8]. Among physical fitness components, cardiorespiratory fitness has attracted most scientific interest with regards to the evaluation of children [2,8,9,10]. Most of the previous research had a cross-sectional design, whereas few studies evaluated physical fitness annually at these very early ages [8].

In addition to physical fitness, nutrition has also been shown to be related to health. For instance, the Mediterranean Diet (MD) has been observed to reduce the risk for cardiovascular and cerebrovascular diseases [11,12]. MD has been widely analyzed in adults [13], adolescents [14,15], and children [16], but not in those aged 3 to 5 years. 

Moreover, the environment of residence (rural versus urban) might also influence the level of physical fitness. Actually, there was information on the effect of the environment in different countries such as the United Kingdom [17], Taiwan [18], Kosovo [19], New Caledonia [20], Croatia [21], and Spain [22,23]. The size of the population in the place of residence might be associated with opportunities for access to sports facilities, occasions for physical activity, lifestyle, or eating habits [15,22]. However, the abovementioned studies have been conducted in adolescents, whereas few information was available for preschool children. So far, the role of place of residence on physical fitness and nutrition has been established for older children, and physical fitness has been related to physical activity and health [1,11]. Moreover, physical fitness and nutrition may be tracked from childhood through adolescence to adulthood. Thus, being aware of the variation of physical fitness between rural and urban environment in preschool children would be of great practical importance for pediatrists, nutritionists, and sport scientists. To the best of our knowledge, no study has ever examined nutritional aspects and the combined effect of environment and age in physical condition of preschool children. To know, in a combined way, the role of the residence, the adherence to the MD, as well as the level of physical fitness, would help to develop optimal exercise interventions from an educational and public health point of view, to avoid risk factors in the future caused by sedentary lifestyle [21]. Therefore, the aim of the present study was to examine the effect of place of residence on physical fitness and adherence in Mediterranean diet (AMD) in 3–5-year-old children, i.e., whether those who live in urban areas differ than those in rural and whether any difference varied by age.

## 2. Materials and Methods

### 2.1. Sample

A total of 363 preschool children (girls, *n* = 167; boys, *n* = 196) of 3 (*n* = 49 and *n* = 56, respectively), 4 (*n* = 61 and *n* = 63, respectively), and 5 years old (*n* = 57 and *n* = 77, respectively) participated in the present study (Table 1). With regards to the place of residence, 242 participants lived in urban and 121 in rural places. The inclusion criteria were as follows: to be between 3 and 5 years old, and have the ability to participate in both regular daily physical activity and physical education. Written informed consent was obtained from parents or guardians of all children prior to their participation. The Ethical Committee of the Local Institution approved this study.

### 2.2. Procedures

Anthropometric characteristics: Body mass, height, and waist circumference were measured. Body mass index (BMI) was calculated as the quotient of BM (kg) to height squared (m^2^). An electronic weight scale (HD-351 Tanita, IL, USA) was used for BM measurement (in the nearest 0.1 kg) and a portable stadiometer (SECA, Leicester, UK) for stature (0.1 cm). Waist circumference was measured using an anthropometric unstretchable tape (Holtain).

Medicine ball throw test (MBTT): Each student sat on the floor with their back against the wall. The student held the ball (1 kg) in front of them with both hands, resting it against their lap. Each participant performed two practice throws, and then the distance of the next two throws was recorded, with a 1 to 2-min rest between each throw. The best measurement was recorded [24].

Flamingo balance test: This test assessed the individual’s postural stability while balancing on one foot with eyes open. The participant, with hands on hips, lifted one leg and attempted to balance for a maximum time of 30 s. The number of imbalances, resulting on jumping on the support leg or free-leg touch down, during the established time, was recorded [25].

Bouncing ball: This test assessed the participants’ bouncing and catching ability with a large ball. The skill required eye-hand coordination, postural stability, body positioning, and control of force over the ball. Standing in an upright position, the participant bounced and caught the ball as many times as possible for a set time of 20 s and the number of complete catches was recorded [25].

25 m sprint: This test assessed running speed and required dynamic balance, coordination of arms and legs, and anaerobic power. Individually, the participants ran a 25-m course at maximal speed. The time recorded was at 5-m and 25-m with photoelectric cells (Smartspeed-Lite, Fusion Sport, Brisbane, Australia). Two runs were completed and the students rested for 20 min between runs. The shorter time of the two runs was recorded in seconds.

Lateral jump: Children performed one trail of 15 s lateral jumping. A box (60 × 96 m) was drawn on the ground. A jump was correct when momentum and reception was with both feet and inside the rectangle [26].

Standing broad jump: The standing broad jump (SBJ) test was used as an indicator of lower limb strength. It consists of jumping the longest distance possible from a standing start (without racing ahead) and swinging both arms. The distance is measured from the take-off line to the point where the back of the heel nearest to the take-off line lands on the ground [27].

Crawling: Duration of time to crawl (on all fours) to and around a 5-m target [28].

Shuttle run test: It consists of running back and forth between two lines 10 m during 3 min. The maximum distance is recorded [2].

Adherence to the Mediterranean Diet: To assess AMD, a self-administered questionnaire (KIDMED) was used [29]. The development of the KIDMED index was based on principles that sustain Mediterranean dietary patterns and undermine it. The questionnaire contains 16 items and the index ranges from 0 to 12. Four questions denote a negative connotation with respect to the Mediterranean diet and assign a value of −1, whereas the rest of the questions have a positive aspect and provide a score of +1. The sums of the values from the administered test were classified into three levels: good (8–12), average (4–7), and poor (0–3) AMD.

### 2.3. Statistical Analysis

Descriptive statistics were calculated for all variables. The student *t*-test was used to examine sex differences and the magnitude of these differences was evaluated by Cohen’s d. A two-way analysis of variance examined, followed by a Bonferroni post-hoc analysis, the main effects of age and residence, and the age–residence interaction on all anthropometric characteristics and physical fitness. The relationship of AMD with sex, age, and residence was examined using chi-square test (χ^2^). The statistical and data analysis was performed using the statistical software IBM SPSS v.23 (IBM, Chicago, IL, USA) and GraphPad Prism v. 7.0 (GraphPad Software, San Diego, CA, USA). Alpha level was set at 0.05.

## 3. Results

The descriptive characteristics of the sample are presented in Table 1.

Figure 1 and Figure 2 show the relation of physical condition by sex, age, and place of residence.

Adherence to Mediterranean diet according to sex, age, and place of residence is shown in Table 2.

### 3.1. Sex Differences

Better performance in ball bouncing (d = 0.23), MBTT (d = 0.38), 25 m sprint (d = −0.33), SBJ (d = 0.37), crawling (d = −0.38), and SRT (d = 0.24) were observed in boys than in girls (*p* ≤ 0.034; Figure 1). No differences were found in age, weight, height, BMI, waist circumference, Flamingo, and lateral jump (*p* ≥ 0.051, d ≤ 0.21; Figure 2). In addition, a sex–AMD association was observed (χ^2^ = 8.910, *p* = 0.012, φ = 0.175) with more boys reporting good AMD than girls (Table 2).

### 3.2. Main Effect of Residence

In girls, a main effect of residence on height (η^2^ = 0.066), lateral jump (η^2^ = 0.033), and SRT (*p* ≤ 0.020, η^2^ = 0.047) was shown, but not on weight, BMI, waist circumference, bouncing ball, Flamingo, MBTT, 25 m sprint, SBJ, and crawling (*p* ≥ 0.123, η^2^ ≤ 0.015). In boys, a main effect of residence on height (η^2^ = 0.029), bouncing ball (η^2^ = 0.022), and SRT (*p* ≤ 0.039, η^2^ = 0.082) was found, but not on weight, BMI, waist circumference, Flamingo, MBTT, 25 m sprint, SBJ, lateral jump, and crawling (*p* ≥ 0.077, η^2^ ≤ 0.016). Furthermore, no place of residence–AMD association was shown (χ^2^ = 2.657, *p* = 0.265, φ = 0.095) with AMD being similar in children living in rural and urban areas.

### 3.3. Main Effect of Age

In girls, a main effect of age on weight (η^2^ = 0.298), height (η^2^ = 0.544), waist circumference (η^2^ = 0.076), bouncing ball (η^2^ = 0.284), Flamingo (η^2^ = 0.280, MBTT (η^2^ = 0.310), 25 m sprint (η^2^ = 0.310), SBJ (η^2^ = 0.475), lateral jump (η^2^ = 0.345), crawling (η^2^ = 0.358), and SRT (*p* ≤ 0.002, η^2^ = 0.164) was shown, but not on BMI (*p* = 0.239, η^2^ = 0.018). In boys, a main effect of age on weight (η^2^ = 0.226), height (η^2^ = 0.547), bouncing ball (η^2^ = 0.214), Flamingo (η^2^ = 0.211), MBTT (η^2^ = 0.325), 25 m sprint (η^2^ = 0.382), SBJ (η^2^ = 0.341), lateral jump (η^2^ = 0.301), crawling (η^2^ = 0.513), and SRT (*p* < 0.001, η^2^ = 0.134) was observed, but not on BMI and waist circumference (*p* ≥ 0.146, η^2^ ≤ 0.020). Moreover, an age–AMD association was observed (χ^2^ = 12.034, *p* = 0.017, φ = 0.203) with relatively more 3-year-old children reporting good AMD than their older counterparts.

### 3.4. Age–Residence Interaction

In girls, an age–residence interaction on MBTT (η^2^ = 0.048), lateral jump (η^2^ = 0.081), and SRT (*p* ≤ 0.018, η^2^ = 0.071) was found, but not on weight, height, BMI, waist circumference, bouncing ball, Flamingo, 25 m sprint, SBJ, and crawling (*p* ≥ 0.085, η^2^ ≤ 0.030). In boys, an age–residence interaction on height (*p* = 0.029, η^2^ = 0.037) was shown, but not on weight, BMI, waist circumference, bouncing ball, Flamingo, MBTT, 25 m sprint, SBJ, lateral jump, crawling, and SRT (*p* ≥ 0.121, η^2^ ≤ 0.022).

## 4. Discussion

The main findings of the present study were that (a) boys had better performance in ball bouncing, MBTT, 25 m sprint, SBJ, crawling, and SRT than girls and the magnitude of these differences was small; (b) preschoolers in urban residence were taller than those in rural and had better performance in SRT; (c) older preschoolers had larger anthropometric characteristics and better performance than younger preschoolers; (d) the magnitude of the effect of age was larger than the effect of residence; and (e) good AMD was more prevalent in boys than in girls and in 3-year-old children than in their older peers, but was not related to place of residence. Under our knowledge, the novelty of this study is to relate the role of residence, MD, and physical condition in a sample with such an early age.

Actually, there was still a lack of strength measurement in preschool children [24]. For instance, trunk muscular resistance tests have not been validated in children from 2 to 6 years old [28]. The present study included tests that measured muscle strength globally in motor patterns as recommended [30]. In the present study, the boys had higher values in ball bouncing, MBTT, 25 m sprint, SBJ, crawling, and SRT than the girls, which was in agreement with previous research [18]. It has been observed that children had better physical fitness levels at 9 years, especially in manual dynamometry [31]. The strength tests developed, such as the MBTT and the SBJ, showed values that were very far from those found in other studies that indicated very similar values between sex in the ages of 3 to 6 years [32]. Based on the previous idea that the differences of the muscle strength fitness component appeared when years were completed [33]. It should be highlighted that the weight and height had influence on muscle strength [24,34]. However, it was necessary to delve into this topic, since it had been observed in Spanish preschool children that there were differences showing higher values in boys than in girls in SBJ [35], highlighting the importance of the place of residence. This trend had been observed in coordination, where boys had better values in ball bouncing and crawling than girls, although there was a lack in scientific studies that indicated the cause.

Also, there were differences in speed velocity test and SRT in boys versus girls (Figure 1 and Figure 2). In this way, there were contradictions in scientific literature, because there were a lot of studies that indicated these differences in children from 5 to 9 years old [18,35,36], whereas other studies did not indicate such differences [5,9,37]. The influence of the place of residence was highlighted once again, since similar data were indicated in countries such as Kenya, Taiwan, and Ecuador [18,35,36] and interestingly, different in countries such as Spain or Switzerland [5,9,37], where the socioeconomic level or lifestyle could have a direct influence on health parameters [22,23,38]. Nonetheless, the parameter that remained to be determined was the one related to cardiorespiratory fitness, since it has been observed that a good level led to a healthier and more active life [10]. There was evidence of how, in children 6 to 10 years old, levels of cardiorespiratory fitness were positively associated with cardiovascular disease risk in only two years [39], so this analysis in preschool age are very important. Also, the same trend was detected with the MD, where children had a more appropriate MD than girls, which should be analyzed in the future, since, at adolescent ages, these differences did not seem to exist [15,16].

This fact came together with the idea of the importance of the place of residence. In general, it was observed that preschoolers in the urban environment had higher values in height and SRT (Figure 1 and Figure 2). These data coincided with studies of the Kenya and Caledonia populations [20,40], but not with previous research in Spain, where children from rural areas had higher anthropometric and physical condition values than those from the urban environment [22,23]. The abovementioned studies included ages of primary and secondary education. On the other hand, there were studies which did not show differences in physical condition between rural and urban population [18,19,36]. Among the physical fitness components, SRT was the only one where boys from rural areas had better results than those from the urban environment, which was in agreement with previous research [36]. It has been shown that boys from rural areas, between 6 and 10 years old, showed higher values than those of urban environment in tests of strength and speed [18,19], which contradicted those found in the present study. Maybe, the inconsistency of these results was due to the lack of conditional studies in children between the ages of 3 and 5 years, being possible that at preschool ages the contextual differences did not clearly mark so many differences in the physical condition. In the same way, there were no differences in the environment regarding the MD level. At adolescent ages, there seemed to be a tendency for urban environments to have less adherence to MD than in rural settings [14,15]. It was possible that the age—since there was greater independence and access to consume nutrients considered “garbage” in the urban environment—influences its occurrence in older population, but it was a finding that should be investigated in the future. 

With regards to the effect of age, the present study showed that the age of 3 to 5 years influenced physical condition as shown by the statistically significant differences in most of the variables (Figure 1 and Figure 2). These data coincided with those found in different articles [5,31,34,35] that indicated an effect of age on the physical and conditional development of preschool children, since they were in the process of development and maturation [41]. Also, one of the most outstanding results of this study was that the age effect had larger magnitude than the environment (rural vs. urban). Therefore, it was recommended to establish strategies aimed at stimulating the physical abilities of children aged between 3 and 5 years [6], to develop pro-active and healthy children and young people, since this age seemed to play a key role in the future development of humans. 

A limitation of the findings was that the physical activity levels and nutritional habits of participants’ families were not considered. It was acknowledged that both physical activity and nutritional habits clustered into families and consisted confounded factors [42]. Strength of the present study was its novelty, since, to the best of our knowledge, it was the first one to examine the variation physical fitness and nutritional habits knowledge by sex, age, and place of residence. This type of studies is necessary in order to generate future interventions based on the reality of the context.

In conclusion, boys outscored girls in most tests, older preschoolers scored better than their younger peers, and differences in anthropometry between rural and urban were identified. In addition, the role of age on physical fitness and nutritional habits is more pronounced than that of place of residence in preschoolers. Therefore, these findings identified the need to develop exercise and nutrition intervention programs in preschoolers considering sex, age, and place of residence.

## Figures and Tables

**Figure 1 nutrients-10-01855-f001:**
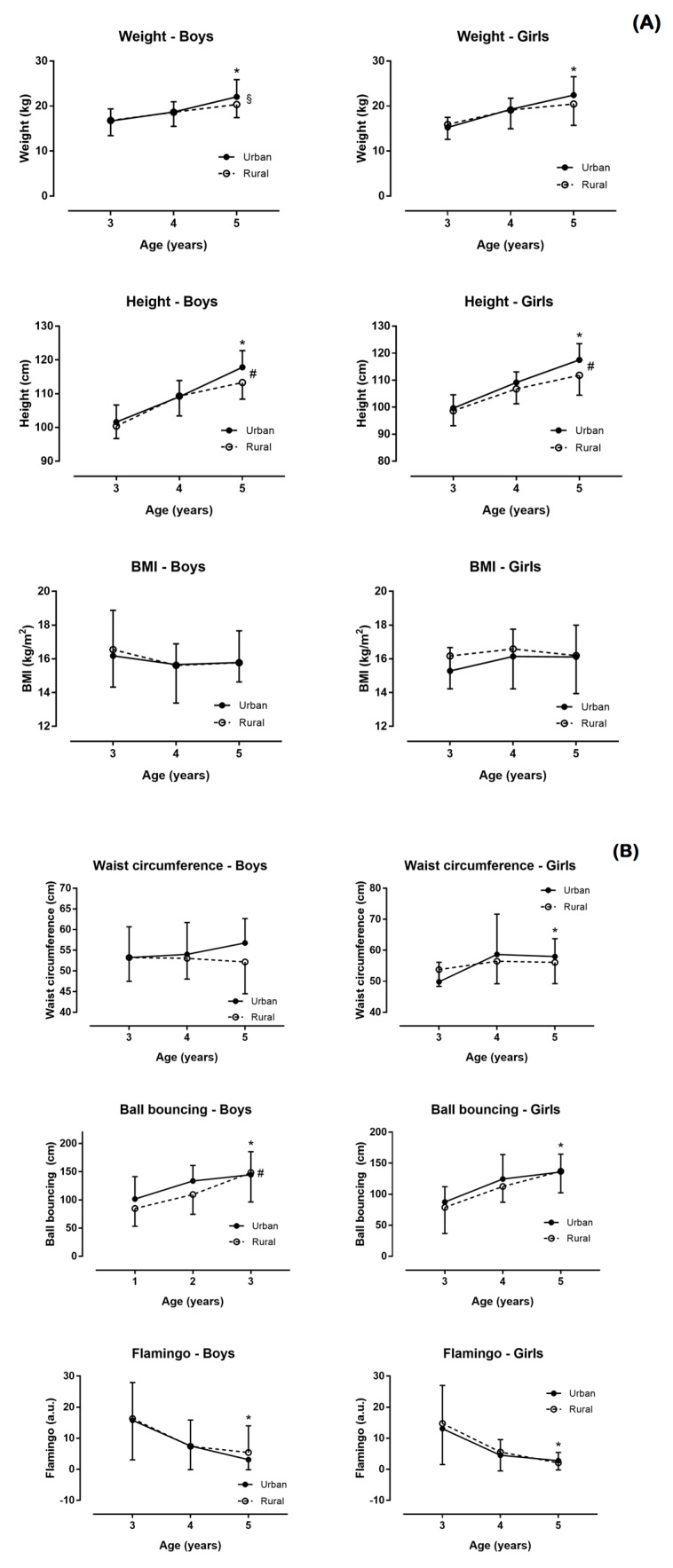
Weight, height, body mass index (BMI) (**A**), waist circumference, ball bouncing, and Flamingo test (**B**) by sex, age, and place of residence. *, age difference; #, difference by place of residence (rural versus urban); §, age × place of residence interaction.

**Figure 2 nutrients-10-01855-f002:**
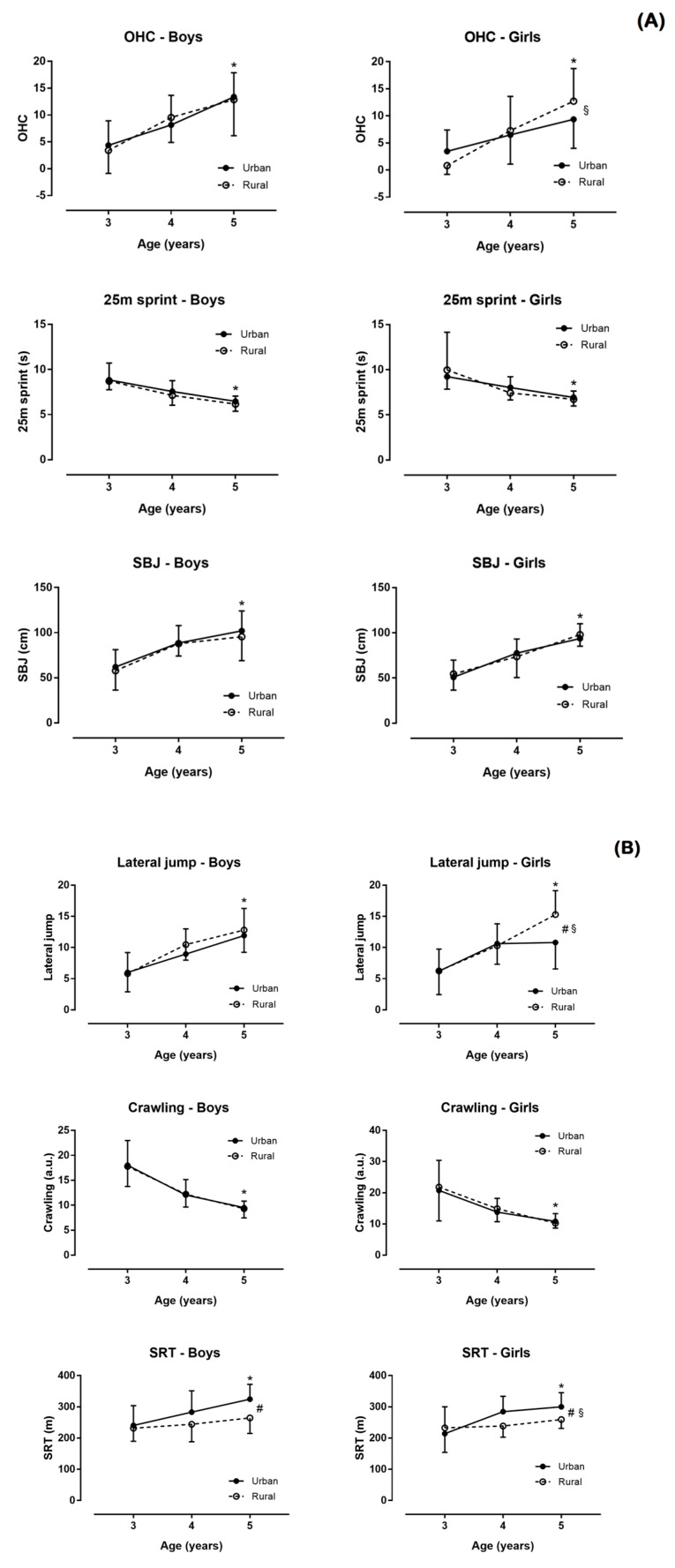
Medicine ball throw test (MBTT), 25 m sprint, standing broad jump (SBJ) (**A**), lateral jump, crawling, and shuttle run test (**B**) by sex, age, and place of residence. *, age difference; #, difference by place of residence (rural versus urban); §, age × place of residence interaction.

**Table 1 nutrients-10-01855-t001:** Descriptive characteristics of participants.

	Girls (*n* = 167)	Boys (*n* = 196)
Age (years)	4.0 ± 0.8	4.1 ± 0.8
Weight (kg)	19.0 ± 4.3	19.2 ± 3.7
Height (cm)	108.1 ± 8.6	109.8 ± 8.1
BMI (kg·m^−2^)	16.0 ± 1.9	15.9 ± 2.0
Waist circumference (cm)	55.7 ± 8.9	54.3 ± 6.8
MBTT (cm)	115 ± 39	125 ± 44
Flamingo (n)	6.7 ± 8.6	8.5 ± 10.0
Ball bouncing (cm)	6.9 ± 6.1	9.2 ± 6.2
25 m sprint (s)	8.0 ± 1.9	7.4 ± 1.5
Lateral jump (n)	9.9 ± 4.6	9.5 ± 4.5
SBJ (cm)	75.6 ± 24.4	85.1 ± 26.1
Crawling (s)	15.0 ± 6.9	12.7 ± 4.7
SRT (m)	260 ± 58	275 ± 66

BMI = body mass index, MBTT = Medicine Ball Throw Test, SBJ = standing broad jump, SRT = shuttle run test.

**Table 2 nutrients-10-01855-t002:** Adherence to Mediterranean diet according to sex, age, and place of residence.

	Adherence to Mediterranean Diet					
	Good (*n*)	Good (%)	Average (*n*)	Average (%)	Poor (*n*)	Poor (%)
Sex						
Girls (*n* = 134)	60	44.8	67	50.0	7	5.2
Boys (*n* = 158)	81	51.3	77	48.7	0	0.0
Age (years)						
3 (*n* = 74)	47	63.5	27	36.5	0	0.0
4 (*n* = 119)	54	45.4	60	50.4	5	4.2
5 (*n* = 98)	40	40.8	56	57.1	2	2.0
Residence						
Urban (*n* = 203)	97	47.8	103	50.7	3	1.5
Rural (*n* = 89)	44	49.4	41	46.1	4	4.5

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
