# Peer review of "The Effect of Place of Residence on Physical Fitness and Adherence to Mediterranean Diet in 3–5-Year-Old Girls and Boys: Urban vs. Rural"

_nutrients, 2018, doi:10.3390/nu10121855_

Round 1

Reviewer 1 Report

 The study by Gema Torres-Luque et al investigates the impact of place of residence on physical fitness and adherence in Mediterranean diet in children in urban and rural populations

The study has been conducted well and the manuscript is well presented. However, there are several issues with the manuscript in its current form that needed to be addressed.

1.     In Introduction should be presented the rationale of such study. It is not clear what is the importance of study. How the obtained results will help us in better understanding of the role of residence, diet and physical activity in childhood.

2.     It is hard to see the novelty of the current study or how it pushed our understanding forward. This should be clearly highlighted in Discussion and Conclusions.

3.     Do the authors have any detail on the underlying mechanism linking place of residence with physical fitness and adherence to Mediterranean diet in children? As it currently stands, the study is very descriptive in nature and it is hard to see how this study moves our understanding along.

4.     In Materials and Methods I would like to see the reason why the participants were not randomized into groups and some information about level of physical activity and type of diet in families which kids are assigned in the study. It is possible that the family physical activity and type of diet is more important that, for example, urban or rural type of population.

5.     There is a number of typos in the manuscript. For example lane 193 “cause5. Conclusions” In case if Conclusion starts from the lane 193 the Authors have to make it shorter and move some stuff to Discussion. It is also mentioned that Summary is in the very last paragraph of manuscript.  Please, make one.

Author Response

The study by Gema Torres-Luque et al investigates the impact of place of residence on physical fitness and adherence in Mediterranean diet in children in urban and rural populations

Dear Review 1:

Thank you very much for your contribution at this paper. We think that your contribution improves the study. We answer you below:

The study has been conducted well and the manuscript is well presented. However, there are several issues with the manuscript in its current form that needed to be addressed.

1.     In Introduction should be presented the rationale of such study. It is not clear what is the importance of study. How the obtained results will help us in better understanding of the role of residence, diet and physical activity in childhood.

We agree with the expert reviewer and added “Aspects such as physical fitness and nutritional habits are of vital importance at an early age of human life, and the place of residence and age, can be decisive [1, 2].” in the beginning of the introduction and “So far, the role of place of residence on physical fitness and nutrition has been established for older children, and physical fitness has been related to physical activity and health [1, 11]. Moreover, physical fitness and nutrition might be tracked from childhood through adolescence to adulthood. Thus, being aware of the variation of physical fitness between rural and urban environment in pre-school children would be of great practical importance for pediatrists, nutritionists and sport scientists.” in the end of the introduction.

2.     It is hard to see the novelty of the current study or how it pushed our understanding forward. This should be clearly highlighted in Discussion and Conclusions.

We agree with the expert reviewer and added this information in the end of Discussion.

3.     Do the authors have any detail on the underlying mechanism linking place of residence with physical fitness and adherence to Mediterranean diet in children? As it currently stands, the study is very descriptive in nature and it is hard to see how this study moves our understanding along.

Maybe the study has a descriptive character, but we considered that is very interesting by the age of the sample. It is not normally in the scientific literature studies with these component interrrelation. We think is a stronger of this study.

In others population (adolescents or adults) exist information about physical condition and place of residence; and for the other hand, MD and place of residence. But, there are a few studies with these components in the same study. In addittion, under our knowledge, there are not in a population by 3 to 5 years.

(e.g. Grao-Cruz et al., Adherence to mediterranean diet in rural urban adolescents of southern spain, life satisfaction, anthropometry, and physical and sedentary activities. Nutrición Hospitalaria 2013, 28(4), 1129-1135. // Chacon -Cuberos et al. Adherence to Mediterranean diet in university students and its relationship with digital leisure habits. Nutr. Hosp. [online]. 2016, vol.33, n.2, pp.405-410. ISSN 1699-5198. // Grigoropoulou et al., Urban environment adherence to the Mediterranean diet and prevalence of asthma symptoms among 10- to 12-year-old children: The Physical Activity, Nutrition, and Allergies in Children Examined in Athens study, 2011. // De la Cruz-Sánchez, E., Aguirre-Gómez, M.D., Pino-Ortega, J., Díaz-Suárez, A., Valero-Valenzuela, A., & García-Pallarés, J. (2013). Rural – urban diferences in children’s physical itness. Revista de Psicología del Deporte, 21(2), 359-363 // Andrade et al., Physical fitness among urban and rural Ecuadorian adolescents and its association with blood lipids: a cross sectional study. BMC Pediatrics, 2014).

4.     In Materials and Methods I would like to see the reason why the participants were not randomized into groups and some information about level of physical activity and type of diet in families which kids are assigned in the study. It is possible that the family physical activity and type of diet is more important that, for example, urban or rural type of population.

We agree with the expert reviewer and added the information about the participants in the methods (“363 preschool children (girls, n=167; boys, n=196) of 3 (n=49 and n=56, respectively), 4 (n= 61 and n=63, respectively) and 5 years old (n=57 and n=77, respectively) participated in the present study (Table 1). With regards to the place of residence, 242 participants lived in urban and 121 in rural place.”). However, no information was available about family’s physical activity and nutrition; since, this is an important aspect we added as a limitation before conclusion (“A limitation of the findings was that the physical activity levels and nutritional habits of participants’ families were not considered. It was acknowledged that both physical activity and nutritional habits clustered into families and consisted confounded factors [42].”).

5.     There is a number of typos in the manuscript. For example lane 193 “cause5. Conclusions” In case if Conclusion starts from the lane 193 the Authors have to make it shorter and move some stuff to Discussion. It is also mentioned that Summary is in the very last paragraph of manuscript.  Please, make one.

Thank you. We have corrected them.

Thank you very much for your job.

Reviewer 2 Report

Dear Editor,

I am pleased to review the assigned manuscript. Overall, a proposed review looks good to me – Authors really did a good job and presented very nice & relevant literature. The English is generally satisfactory, although there are some places where corrections and/or changes are required. The study does have few minor issues. The authors should improve the paper following these suggestions.

Abstract need bit of attention and should cover theme of whole manuscript

The introduction should be better organised. Some of the sentences are not well structured, should be clarified and rewritten. It is advice to link the story in a better way in an introduction to convey a proper message to readers.

Material and Method section is good.

Result section is written well but looks messy because of lot of P values, I suggest could you please add one table for P-Values and try, it’s just suggestion if it look better

I would suggest authors to improve the graphs – they are on very small scale, it’s great data you can try to split and use a,b,c legends.

Discussion is pretty good and well justified.

Please recheck the reference style, some of the references are not according to the journal instructions

Author Response

Dear review 2.

Thank you very much for your contribution at this paper. We think that your contribution improves the study. We answer you below:

Dear Editor,

I am pleased to review the assigned manuscript. Overall, a proposed review looks good to me – Authors really did a good job and presented very nice & relevant literature. The English is generally satisfactory, although there are some places where corrections and/or changes are required. The study does have few minor issues. The authors should improve the paper following these suggestions.

Abstract need bit of attention and should cover theme of whole manuscript.

Thank you. We have done it

The introduction should be better organised. Some of the sentences are not well structured, should be clarified and rewritten. It is advice to link the story in a better way in an introduction to convey a proper message to readers.

Thank you. We have done it

Material and Method section is good.

Result section is written well but looks messy because of lot of P values, I suggest could you please add one table for P-Values and try, it’s just suggestion if it look better

We agree with the expert reviewer that this part needed improvement. Instead of a table, we remained a representative p value in the end of the relevant sentences so they look better. This modification did not influence the presentation of the magnitude of differences, as we maintained the effect sizes.

I would suggest authors to improve the graphs – they are on very small scale, it’s great data you can try to split and use a,b,c legends.

We agree with the expert reviewer and improved the graphs by increasing their size and splitting them as suggested.

Discussion is pretty good and well justified.

Please recheck the reference style, some of the references are not according to the journal instructions

Thank you. We have done it

Thank you very much for your job.

Round 2

Reviewer 1 Report

Manuscript looks good for me. Good luck!